# Intelligent Control of Swarm Robotics Employing Biomimetic Deep Learning

Haoxiang Zhang [1] and Lei Liu [1,2,*]

1   School of Optical-Electrical, University of Shanghai for Science and Technology, Shanghai 200093, China; 192550420@st.usst.edu.cn
2   Business School, University of Shanghai for Science and Technology, Shanghai 200093, China
*   Correspondence: liulei@usst.edu.cn

**Abstract:** The collective motion of biological species has robust and flexible characteristics. Since the individual of the biological group interacts with other neighbors asymmetrically, which means the pairwise interaction presents asymmetrical characteristics during the collective motion, building the model of the pairwise interaction of the individual is still full of challenges. Based on deep learning (DL) technology, experimental data of the collective motion on *Hemigrammus rhodostomus* fish are analyzed to build an individual interaction model with multi-parameter input. First, a Deep Neural Network (DNN) structure for pairwise interaction is designed. Then, the interaction model is obtained by means of DNN proper training. We propose a novel key neighbor selection strategy, which is called the Largest Visual Pressure Selection (LVPS) method, to deal with multi-neighbor interaction. Based on the information of the key neighbor identified by LVPS, the individual uses the properly trained DNN model for the pairwise interaction. Compared with other key neighbor selection strategies, the statistical properties of the collective motion simulated by our proposed DNN model are more consistent with those of fish experiments. The simulation shows that our proposed method can extend to large-scale group collective motion for aggregation control. Thereby, the individual can take advantage of quite limited local information to collaboratively achieve large-scale collective motion. Finally, we demonstrate swarm robotics collective motion in an experimental platform. The proposed control method is simple to use, applicable for different scales, and fast for calculation. Thus, it has broad application prospects in the fields of multi-robotics control, intelligent transportation systems, saturated cluster attacks, and multi-agent logistics, among other fields.

**Keywords:** collective motion; swarm robotics; deep neural network; intelligent control; self-organization

## 1. Introduction

Collective motion occurs widely in group-living animals, which can help the groups to adapt to the environment through solving problems collectively, such as by predator avoidance or cluster foraging. Hence, extensive research has been focusing on this topic. For instance, the collective motion of desert locusts was studied in [1], which revealed the principle of the insect's aggregation. Cavagna took advantage of the machine vision technology to capture the trajectory data of the large-scale purple-winged pheasant and built a model of social interaction for collective motion [2]. Altshuler et al. [3], inspired by Helbing [4], discovered an interesting collective behavior of ant colonies: the symmetrical breaking of collective escape motion. These works show that most collective motions are mainly caused by the social interactions between individuals [5,6].

Social interaction is defined as the transmission and processing of distributed information by the individual, which can be divided into two main structures: hierarchical structure and egalitarian structure [7]. Most of the mammalian beasts in nature are organized in hierarchical structures. On the contrary, the collective motion of bacteria is often supposed as egalitarian structures. Furthermore, the bird flocks and fish schools

are basically between hierarchical structures and egalitarian structures. The hierarchical structure of interaction, such as leader–follower relationships in the group, is more ubiquitous, which can lead to more effective organization [8,9]. Therefore, after the leadership literature of Couzin et al. [10], the hierarchical interaction model has attracted more and more attention.

However, it is difficult to study hierarchical interaction. First, the pairwise interaction between two individuals should be treated asymmetrically, which leads to different models for two paired individuals. Second, the input of most existing interaction models is based on the information of the position of the individual and its neighbors [11]. However, the literature [12] proposes that collective motion should also be related to the speed value of each individual. In fact, there are many parameters of each individual for potential explanation of the interaction. However, it is difficult to build mathematically analytical models with both relative position and speed inputs for explaining the asymmetrical interaction [11]. Therefore, building a multi-parameter model of distributed interaction for large-scale collective behavior is still an open and challenging problem.

The data-driven models, especially those based upon Deep Neural Network (DNN), have a strong ability to reveal the relationship between multi-parameter inputs and the individual decision. Because the deep learning (DL) technology is suitable for solving complex mapping problems, and in recent years, DNN models have been widely applied in pattern recognition [13–16], behavior prediction [17,18], and collective motion analysis [19,20]. However, there are few cases of using DNN models to control swarm robotics.

Swarm robotics has many important potential applications, such as nanoparticles controlled by a magnetic field, which can be used for medical treatments [21–24]. It is very difficult to control swarm robotics to formulate collective motion. Most methods for swarm robotics control rely on the control theories [25–30], but the performance of these methods exhibits a lack of flexibility. Thus, Vasarhelyi G. et al. successfully used a social interaction model to realize 30 drones' flexible collective motion [31]. Inspired by them, we used the data-driven method on real fish data to build a social interaction model, which could be used to drive our self-made swarm robotics (Cuboid) to move collectively [32]. However, the above two works never used the DNN interaction model.

Due to the computational limitation of the individual, very few works have explicitly addressed the question about how the individual sparsely integrates pairwise interaction with all its neighbors in an animal group [33]. Instead of using average contributions of all neighbors, as many models previously proposed [34–39], our previous work suggests that an individual fish pays attention to a few neighbors [32]. This mechanism has the advantage of overcoming the natural limitation of individual information processing [40].

The contribution of this work is listed as follows. First, we use DL technology to build a pairwise interaction model and analyze the pairwise interaction between real fish. Then, we integrate this pairwise model into the collective motion control of multi-agents in different scales. Most researchers argue that each individual should take advantage of the information of all neighbors in a certain domain around the focal individual [11,32], such as the Aoki model [34], the Couzin model [35], and the Vicsek model [36]. Vicsek believes that each individual makes directive decisions with all individuals in a certain range. However, in starling flocks [37,38], it is believed that the motion decision of each individual in collective motion depends only on a limited number of neighbors. We try to reveal that if an individual only interacts with one key neighbor, it can also formulate collective motion through our proposed model. The key neighbor is selected by our algorithm based on the visual information of the focal individual. We name this algorithm the Largest Visual Pressure Selection (LVPS) strategy. This assumption of only one neighbor's attention can significantly reduce the computation load of the individual [32]. The fish experiment is used to verify the similarity of our method's simulation. Finally, we extend our method to the collective motion control of large-scale multi-agents and swarm robotics.

## 2. Materials and Methods

### 2.1. Experimental Data

The fish (*Hemigrammus rhodostomus*) collective motion experimental data were downloaded in the supplementary materials of [11,32]. These data were extracted from experimental video via idTracker software [41] (see Figure 1A). We used 30-mm long and 2.5-mm wide fish for the collective motion experiment. The fish had a burst-and-coast swimming pattern, which means that the fish first turn their direction with speed increasing and then decelerate for a straight slide [11]. Most heading angle change takes place entirely at the beginning of the acceleration phase. We call the speed increase of fish a "kick." At each time of the kick, the fish makes a motion decision, such as moving distance, kick duration, and heading variation. We detected 60,312 kicks for the five-fish experiment and 147,776 kicks for the two-fish experiment. We decided to use the two-fish experimental data to build our DNN model and the five-fish data to verify the DNN model.

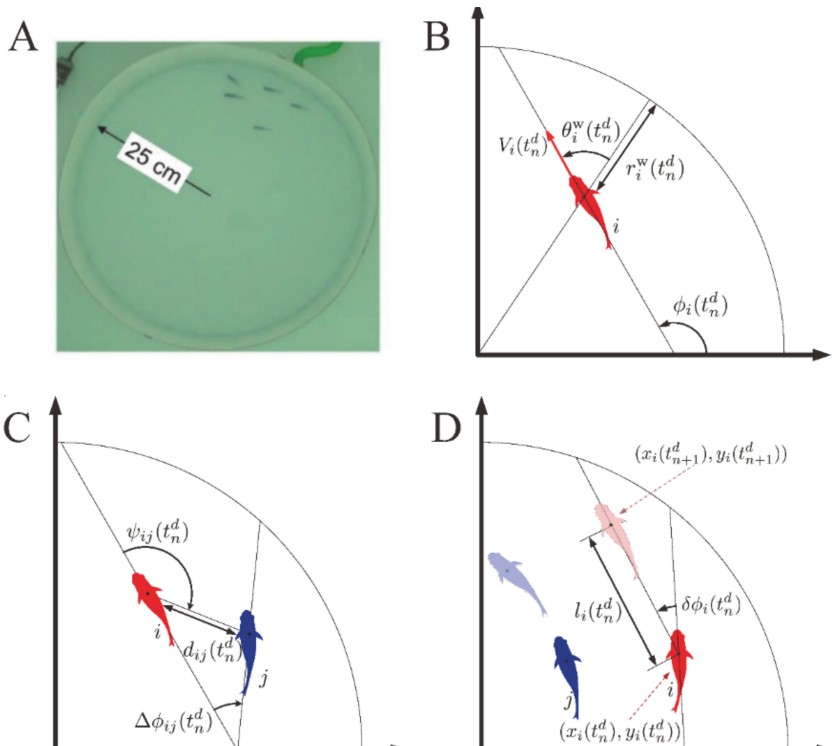

**Figure 1.** The configuration of a real fish experiment and parameters in the global reference systems at the decision time $t_n^d$ (**A**) Five–fish schooling experiment in a tank which radius $R_W = 250$ mm; (**B**) The position parameters of fish, where $r_i^W$ represents the radial distance from fish $i$ to the wall. $\phi_i$ is the heading angle of fish $i$, $V_i$ is the speed of fish $i$, the direction of $V_i$ is determined by the $\phi_i$; $\theta_i^W$ represents the relative angle from fish $i$ to the wall; (**C**) The measurement parameters of neighbor fish (blue). $d_{ij}$ represents the relative distance between fish $i$ and its neighbor $j$; $\psi_{ij}$ represents the angle of fish $i$ perceiving its neighbor $j$ (note that $\psi_{ji} \neq \psi_{ij}$); $\Delta\phi_{ij}(t) = \phi_j(t) - \phi_i(t)$ represents the difference of heading angle between fish $i$ and its neighbor $j$; (**D**) The moving decision parameters of the focal fish (red). $\delta\phi_i(t_n^d)$ is the change of heading angle of fish $i$ at the decision time $t_n^d$, $l_i(t_n^d)$ is the one kick distance between $t_n^d$ and $t_{n+1}^d$.

The heading angle of fish $i$ is described by the angle between the fish velocity vector $\vec{v} = (v_x, v_y)$ and the horizontal line of the x coordinate:

$$\phi_i(t) = \arctan(v_i^y(t)/v_i^x(t)) \tag{1}$$

The positive direction of the fish heading angle is counterclockwise. This heading angle value is restricted to the range of $(-\pi, \pi]$. The fish velocities are calculated by $v_i^x(t) = (x_i(t) - x_i(t - \Delta t))/\Delta t$ and $v_i^y(t) = (y_i(t) - y_i(t - \Delta t))/\Delta t$, where $\vec{p}_i = (x_i(t), y_i(t))$ is the position of the focal fish $i$ and $\Delta t$ is 0.04 s. Based on the position $(x_i(t), y_i(t))$, heading angle $\phi_i(t)$ of the fish $i$, and the radius of the circular tank $R_w$, we can calculate the following local information of the focal fish $i$ with respect to the environment and its neighbors (see Figure 1B–D).

The relative orientation of the focal fish with respect to its neighbor $j$:

The distance to the wall may be described as follows:

$$r_i^W(t) = R_w - \sqrt{x_i(t)^2 + y_i(t)^2} \tag{2}$$

The angle to the wall may be described as follows:

$$\theta_i^W(t) = \phi_i(t) - \arctan(y_i(t)/x_i(t)) \tag{3}$$

The speed of the focal fish $i$ may be described as follows:

$$V_i(t) = \sqrt{v_i^x(t)^2 + v_i^y(t)^2} \tag{4}$$

The distance to the neighbor $j$ may be described as follows:

$$d_{ij}(t) = \sqrt{(x_i(t) - x_j(t))^2 + (y_i(t) - y_j(t))^2} \tag{5}$$

The viewing angle of the neighbor $j$ may be described as follows:

$$\psi_{ij}(t) = \arctan\left( \frac{(x_i(t) - x_j(t))\sin(\phi_i(t)) + ((y_j(t) - y_i(t)))\cos(\phi_i(t))}{(x_j(t) - x_i(t))\cos(\phi_i(t)) + ((y_j(t) - y_i(t)))\sin(\phi_i(t))} \right) \tag{6}$$

The orientation difference from the focal fish to its neighbor $j$ may be described as follows:

$$\Delta\phi_{ij}(t) = \phi_j(t) - \phi_i(t) \tag{7}$$

Since the two positions of one neighbor at two consecutive decision moments can reflect the neighbor position changing with respect to the focal fish, we define the following formula as the average relative speed of neighbor $j$:

$$\Delta V_{ij}(t_n^d) = \left( d_{ij}(t_n^d) - d_{ij}(t_{n-1}^d) \right)/(t_n^d - t_{n-1}^d) \tag{8}$$

Hence, the average speed of the focal fish $i$ can also be defined as

$$V_i(t_n^d) = \sqrt{(x_i(t_n^d) - x_i(t_{n-1}^d))^2 + (y_i(t_n^d) - y_i(t_{n-1}^d))^2}/(t_n^d - t_{n-1}^d) \tag{9}$$

The motion decisions of the focal fish $i$ are the heading changing angle $\delta\phi_i$, the kick distance $l_i$, and the kick duration $KT_i$, which can be calculated by the positions and heading angles of two sequential kicks as follows:

$$\delta\phi_i(t_n^d) = \phi_i(t_{n+1}^d) - \phi_i(t_n^d), \tag{10}$$

$$l_i(t_n^d) = \sqrt{(x_i(t_{n+1}^d) - x_i(t_n^d))^2 + (y_i(t_{n+1}^d) - y_i(t_n^d))^2}, \tag{11}$$

$$KT_i(t_n^d) = t_{n+1}^d - t_n^d, \tag{12}$$

where $t_n^d$ is the decision time when one kick occurs and $t_{n+1}^d$ is the decision time of the next kick.

### 2.2. Deep Neural Network (DNN) Model

Based on the trajectory of the two-fish experiment, a data-driven model for focal fish interaction with respect to the environment and its neighbor was trained. The interaction model could be represented by a function mapping from local information to the motion decision. The information about the environment (25-cm radius circular wall in Figure 1A) could be regarded as a static obstacle. Meanwhile, the moving neighbor could be addressed as the dynamic obstacle. Thus, the focal fish should have taken both types of different information into account to determine the action for the next kick. Due to the large amount of perceptual information for the decision, using mathematical analytical functions to build the interaction model was difficult. Hence, we took advantage of a DNN model to solve this problem.

For the focal fish $i$ at decision time $t_n^d$, (1) the static information includes the distance from the wall $r_i^W(t_n^d)$, the direction of the angle to the wall $\theta_i^W(t_n^d)$, and the average speed of the focal fish $V_i(t_n^d)$ (see Figure 1B); (2) the dynamic information of neighbor $j$ includes the distance to the neighbor $d_{ij}(t_n^d)$, the viewing angle of the neighbor $\psi_{ij}(t_n^d)$, the relative orientation angle with the neighbor $\Delta\phi_{ij}(t_n^d)$, and the average relative speed with the neighbor $\Delta V_{ij}(t_n^d)$ (see Figure 1C); and (3) the above perceptual information is for neural network input. On the other hand, the outputs of the neural network consist of three parts, which are the heading changing angle $\delta\phi_i(t_n^d)$, the kick length $l_i(t_n^d)$, and the duration time of the kick $KT_i(t_n^d)$ (see Figure 1D).

The distribution of the input and output data of the 2-fish experiment is illustrated in Figure 2. It is obvious that the individual always swam near the wall, referring to the distribution of the relative distance from the wall $r_i^W(t_n^d)$ (Figure 2A) and the direction angle to the wall $\theta_i^W(t_n^d)$ (Figure 2B). The social interaction between the two fish was asymmetrical. This means that there existed a temporary leader-and-follower relationship between the two fish. Hence, the peaks of the probability density function (PDF) of the viewing angle of the neighbor $\psi_{ij}(t_n^d)$ (Figure 2E) was close to zero (for the follower) and $\pm\pi$ (for the leader). One fish was always aligned with and close to another (see the PDF of $d_{ij}(t_n^d)$ (Figure 2D), $\Delta V_{ij}(t_n^d)$ (Figure 2G), and $\Delta\phi_{ij}(t_n^d)$ (Figure 2F)). The PDF of $V_i(t_n^d)$ (Figure 2C) shows that the average speed of an individual was about 120 mm/s. Since the sample period of the fish experimental camera was 0.04 s, the PDF of the duration time of the kick $KT_i(t_n^d)$ (Figure 2L) was discrete. The average value of the kick length $l_i(t_n^d)$ (Figure 2K) was about 60 mm (2 body lengths). The high frequency value of the heading changing angle $\delta\phi_i(t_n^d)$ (Figure 2J) was about 20 degrees. Due to the heading angle changing, the viewing angle of the neighbor and the angle to the wall became $\psi_{ij}(t_n^d) + \delta\phi_i(t_n^d)$ and $\theta_i^W(t_n^d) + \delta\phi_i(t_n^d)$, respectively (see Figure 2H,I). We used these data as the record set to train our DNN model for pairwise interaction.

Due to the burst-and-coast motion type, at time $t_n^d$, the decision-making is divided into two sequential phases. The first phase is to determine the heading changing angle $\delta\phi_i(t_n^d)$, because the heading angle changing occurs exactly before fish acceleration. After that, the second phase decision includes the kick length $l_i(t_n^d)$ and kick duration $KT_i(t_n^d)$. These values are then generated based on the new heading angle value $\phi_i(t_n^d) + \delta\phi_i(t_n^d)$. $l_i(t_n^d)$ and $KT_i(t_n^d)$ contain the relative position and average speed information for the suddenly increased speed (kick) and the following passive gliding period. Therefore, we designed two DNN models to mimic the above two decision phases (see Figure 3). The first DNN model was named the Angle Changing Network

The second one was named the Length and Duration Network (LDN). Since the functionality and information input of the two DNN models were similar, the input layer and the hidden layer of the ACN and LDN had the same structure. They both had 7 neurons as an input layer and 3 hidden layers with 10, 20, 50, 20, and 10 neurons. The Rectified Linear Unit (RELU) was set as the activation function for all layers except the output layer. Because the output of the ACN model was a continuous value with a range $(-\pi, \pi]$, the activation function of the output layer of both networks was selected as a Linearly Activated Function, which is suitable for regression applications. (ACN).

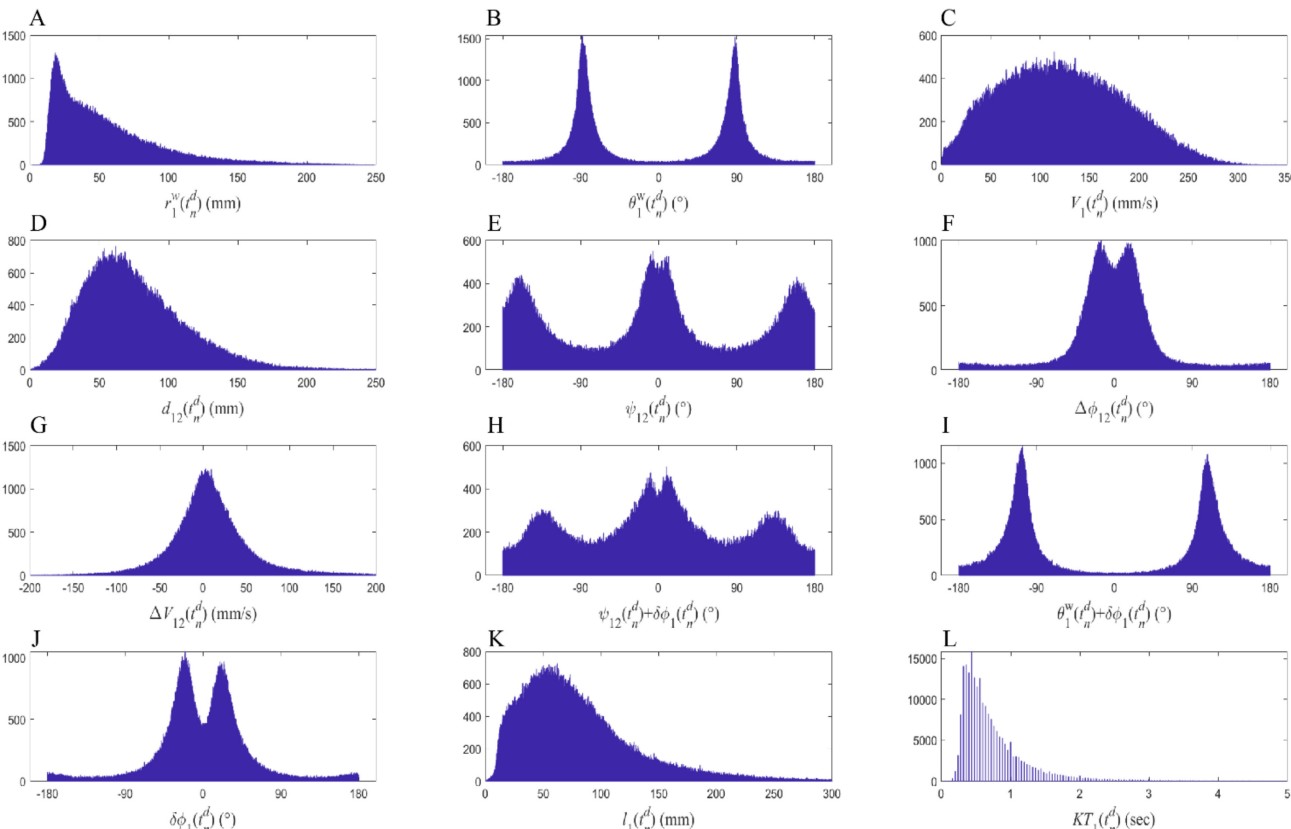

**Figure 2.** The histogram of the measurement value and decision value of the two–fish experiment. For local information of focal fish $i$ at decision time $t_n^d$, the static information includes: (**A**) the distance to the wall $r_i^W(t_n^d)$, (**B**) the direction angle to the wall $\theta_i^W(t_n^d)$ and (**C**) the average speed of the focal fish $V_i(t_n^d)$. While, the dynamic information of neighbor $j$ includes (**D**) the distance to the neighbor $d_{ij}(t_n^d)$, (**E**) the viewing angle of neighbor $\psi_{ij}(t_n^d)$, (**F**) relative orientation angle to the neighbor $\Delta\phi_{ij}(t_n^d)$, and (**G**) the average relative speed with the neighbor $\triangle V_{ij}(t_n^d)$. (**H**) The viewing angle of neighbor $\psi_{ij}(t_n^d) + \delta\phi_i(t_n^d)$ after heading changing of the focal fish. (**I**) The angle to the wall $\theta_i^W(t_n^d) + \delta\phi_i(t_n^d)$ after heading changing of the focal fish. The decision of the focal fish consists of three parts, which are (**J**) the heading changing angle $\delta\phi_i(t_n^d)$, (**K**) the kick length $l_i(t_n^d)$ and (**L**) the duration time of the kick $KT_i(t_n^d)$.

The output layer of the ACN has only one neuron for heading angle change generation. Consider the static information of focal fish $i$ $S_i^{\text{ACN}} = [\ r_i^W(t_n^d) \quad \theta_i^W(t_n^d) \quad V_i(t_n^d)\ ]^T$ and the dynamic information with respect to its neighbor $j$ $D_{ij}^{\text{ACN}} = [\ d_{ij}(t_n^d) \quad \psi_{ij}(t_n^d) \quad \Delta\phi_{ij}(t_n^d) \quad \Delta V_{ij}(t_n^d)\ ]$ such that the output of the ACN is listed as follows:

$$\delta\phi_i(t_n^d) = L\left(f\left(\sum\left(w_{\text{ACN}} \times [S_i^{\text{ACN}}; D_{ij}^{\text{ACN}}]\right)\right)\right) \tag{13}$$

where $w_{\text{ACN}}$ is the weight, $L(\cdot)$ is the Linearly Activated Function, and $f(\cdot)$ is the RELU function. After the ACN generates the output $\delta\phi_i(t_n^d)$, the LDN takes advantage of this value to update its input information. Thus, the static information then becomes $S_i^{\text{LDN}} = [\ r_i^W(t_n^d) \quad \theta_i^W(t_n^d) + \delta\phi_i(t_n^d) \quad V_i(t_n^d)\ ]^T$, which means that the orientation angle to the wall has been changed and impacts the straight motion. For the same reason, the dynamic information of neighbor $j$ for the LDN input is also changed as follows:

$$D_{ij}^{\text{LDN}} = [\ d_{ij}(t_n^d) \quad \psi_{ij}(t_n^d) + \delta\phi_i(t_n^d) \quad \Delta\phi_{ij}(t_n^d) + \delta\phi_i(t_n^d) \quad \Delta V_{ij}(t_n^d)\ ] \tag{14}$$

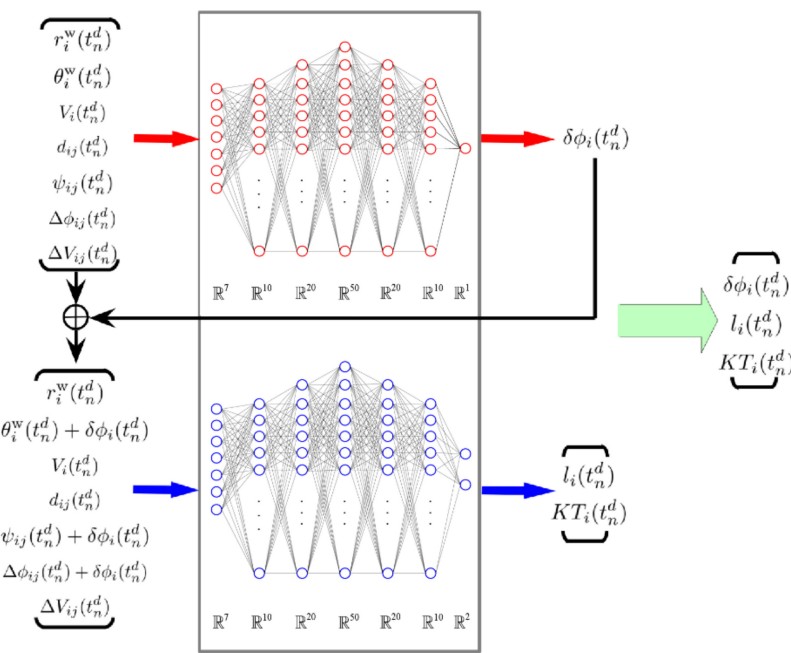

**Figure 3.** The structure of the pairwise interaction network. An Angle Changing Network (ACN) is laid on the top, where it receives the original environmental and the neighbor's information. The outputs are the heading angle changing value $\delta\phi_i$ of the focal fish at decision time $t_n^d$. The Length and Duration Network (LDN) for a straight-moving decision is laid at the bottom. The input of the LDN is the same as the ACN, but the angle value inputs ($\theta_i^W, \psi_{ij}, \Delta\phi_{ij}$) are changed into ($\theta_i^W + \delta\phi_i$, $\psi_{ij} + \delta\phi_i$, $\Delta\phi_{ij} + \delta\phi_i$) because of the heading angle change of the focal fish. The structures of the LDN input layer and hidden layer are similar to those of the ACN. However, the output layer of the LDN includes two neurons, which are the kick length $l_i(t_n^d)$ and the duration time of the kick $KT_i(t_n^d)$.

Hence, the two outputs of the LDN neural network are listed as follows:

$$\begin{bmatrix} l_i(t_n^d) \\ KT_i(t_n^d) \end{bmatrix} = L\left(f\left(\sum\left(w_{\text{LDN}} \times [S_i^{\text{LDN}}; D_{ij}^{\text{LDN}}]\right)\right)\right) \tag{15}$$

where $w_{\text{LDN}}$ represents the weight parameters of the LDN, $L(\cdot)$ is the Linearly Activated Function, and $f(\cdot)$ is the RELU function.

According to the standard procedure of the regression DNN training, we designed the following mean square error formula as a cumulative loss function for both the ACN and LDN:

$$Loss = \frac{1}{N}\sum_{i=1}^{N}(O - \hat{O})^2 \tag{16}$$

where $O$ is the label value of the record data and $\hat{O}$ is the predicted value of the DNN model. $N$ is the number of training samples, which was 147,776 (kicks) for the 2-fish experiment. The label values of the ACN and LDN are $\delta\phi_i(t_n^d)$ and $\begin{bmatrix} l_i(t_n^d) & KT_i(t_n^d) \end{bmatrix}^{\text{T}}, t_n^d \in T^d$, respectively, where $T^d$ is the set for all decision times. We employed the Adam Optimizer [42] to minimize the loss function. The learning rate was set at 0.0005. We randomly selected 20% of all record samples as the test set. The dropout algorithm [43] was adopted to improve the generalization ability of the algorithm.

### 2.3. The Fusion Method of Pairwise Interaction for the Multi-Agents

Instead of using the average contributions of all neighbors as many models previously proposed [34–39], our previous work suggests that an individual paying attention to only a few neighbors can lead to collective motion [32]. This mechanism may overcome the

natural limitation of information each individual can process [40]. In this paper, we wanted to test whether collective motion could emerge from the group when the individual only interacted with one neighbor. Hence, we tested three different neighbor selection strategies to investigate their impacts on collective motion. We then compared the results of a five-agent simulation implemented by different neighbor selection strategies with five real fish experiments. The three neighbor selection strategies were Nearest Neighbor Selection (NNS), Random Neighbor Selection (RNS), and Largest Visual Pressure Selection (LVPS).

For NNS, each individual only considers the information of the nearest neighbor at the decision time $t_n^d$. This means that the individual $i$ only selects neighbor $j$ as its leader, whose distance $d_{ij}(t_n^d)$ is the smallest.

For RNS, the focal fish chooses one neighbor as its leader randomly at the decision time $t_n^d$.

For LVPS, the focal fish chooses its leader as the one with the largest visual pressure. Due to the importance of the fish's vision, the social interaction based on visual sensory input has been intensely researched [44]. Here, we defined the visual pressure of the focal fish as the visual angle of the focal fish with respect to its neighbors (see Figure 4). The larger the visual angle of the neighbor, the greater the visual pressure pressed on the focal fish by this neighbor.

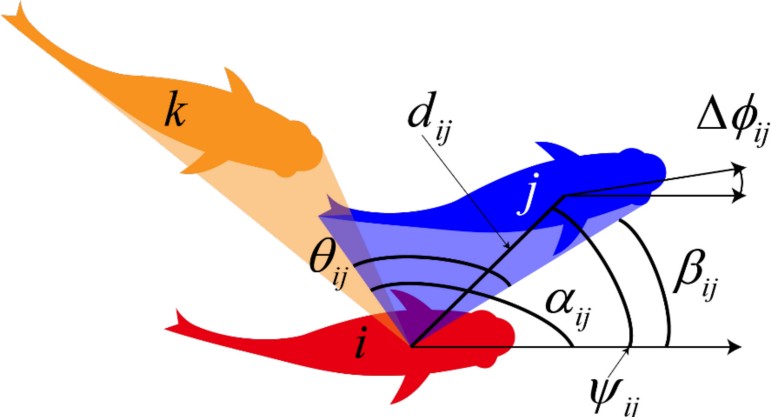

**Figure 4.** Neighbor selection strategy based on visual pressure.

In order to simplify the calculation of the visual pressure angle, one can consider each neighbor as a straight vector. In Figure 4, the red fish $i$ is the focal fish. The blue fish $j$ and yellow fish $k$ are its neighbors. A relative coordinate system is established at the center of the focal fish body. $\alpha_{ij}$ and $\beta_{ij}$ are the visual ending and starting angles of the neighbor $j$, respectively, which are calculated by the following formulas:

$$\alpha_{ij} = \arctan \frac{d_{ij} \times \sin \psi_{ij} + \frac{BL}{2} \times \sin \Delta \phi_{ij}}{d_{ij} \times \cos \psi_{ij} + \frac{BL}{2} \times \cos \Delta \phi_{ij}} \tag{17}$$

$$\beta_{ij} = \arctan \frac{d_{ij} \times \sin \psi_{ij} - \frac{BL}{2} \times \sin \Delta \phi_{ij}}{d_{ij} \times \cos \psi_{ij} - \frac{BL}{2} \times \cos \Delta \phi_{ij}} \tag{18}$$

where $BL$ is the average body length of the fish (30 mm for the fish experiment) and $[d_{ij}, \psi_{ij}, \Delta \phi_{ij}]$ is the local information of fish $i$ with respect to the neighbor $j$. Based on $\alpha_{ij}$ and $\beta_{ij}$, the visual pressure angle $\theta_{ij}$ of neighbor $j$ is then calculated as follows:

$$\theta_{ij} = |\alpha_{ij} - \beta_{ij}| \tag{19}$$

Note that the visual pressure angles of the blue fish $j$ and yellow fish $k$ are overlapped. Hence, it seems that the visual pressure angle of the blue fish should reduce this overlapped part. However, the fish is an intelligent species with imagination and memory. For instance,

fish are able to predict the behavior of short-term hidden prey [45]. Owing to this reason, in spite of only seeing some parts of the neighbor, the focal fish should have the ability to detect the full body of its neighbor. Thus, we tended to use the visual pressure angle with respect to the full body of the neighbors, which is different from the method in the literature [44].

### 2.4. Software Configuration of the Simulation Platform

The DNN interaction model was a core module of the simulation, which was written in the Python and LabVIEW computer languages. The DNN training software was written with TensorFlow, which is a module of Python. Meanwhile, the multi-agent simulation and Graphical User Interface (GUI) were written with LabVIEW because it was convenient for designing Object-Oriented Programming (OOP). OOP is a powerful available programming tool that can easily keep separate the information about each agent in a single software unit. With this facility, all agents in the simulation software are organized by simulation time. At each decision time, the focal agent asks Python for the new motion decision independently. The communication interface between LabVIEW and Python is a client–server program. The server program runs on Python. It receives the DNN input from the focal agent running in the LabVIEW simulation software. The input includes the local static and dynamic information of a focal agent. After neighbor selection for interaction, the Python server program uses TensorFlow to compute the motion decision output of the DNN model (ACN and LDN). Then, this output is downloaded to the LabVIEW client, which sends this decision to the focal agent (see Figure 5).

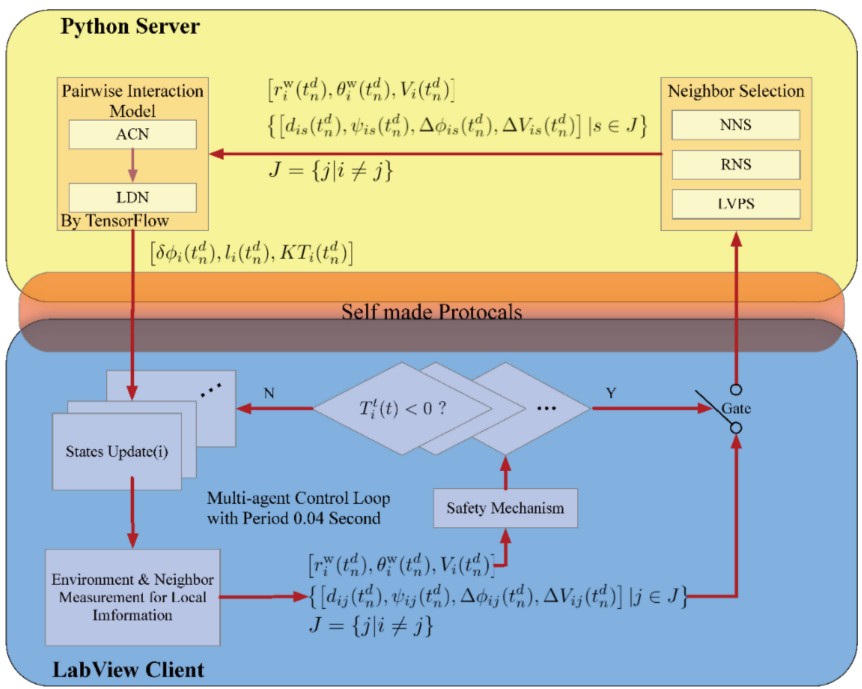

**Figure 5.** The structure of the simulation platform.

LabVIEW updated all the states of the agents according to the simulation clock. The period of the simulation clock $\Delta t$ was 0.04 s. In the global coordinate, the state of agent $i$ is denoted as $[x_i(t), y_i(t), \phi_i(t)]^T$. At decision time $t_n^d$, agent $i$ sends its local environmental information $S_i = [\ r_i^W(t_n^d) \quad \theta_i^W(t_n^d) \quad V_i(t_n^d)\ ]^T$ and all neighbors' information $D_{ij} = [\ d_{ij}(t_n^d) \quad \psi_{ij}(t_n^d) \quad \Delta\phi_{ij}(t_n^d) \quad \Delta V_{ij}(t_n^d)\ ]^T, j \in J$, where $J$ is the set of all the indexes of all neighbors of the focal agent $i$, to the Python server. After the neighbor selection for interaction, TensorFlow computes the decision output and downloads the result $[\ \delta\phi_i(t_n^d) \quad l_i(t_n^d) \quad KT_i(t_n^d)\ ]^T$ to the agent $i$. Then, LabVIEW sets the timer $T_i^t(t)$ of

agent $i$ to the new value $KT_i(t_n^d)$. Meanwhile, the heading angle $\phi_i(t_n^d)$ of agent $i$ $\phi_i$ is set to $\phi_i(t_n^d) + \delta\phi_i(t_n^d)$. After that, LabVIEW updates the state of agent $i$ at every time step with the period $\Delta t$ according to the following formula until $T_i^t(t) < 0$:

$$
\begin{bmatrix} x_i(t+\Delta t) \\ y_i(t+\Delta t) \\ T_i^t(t+\Delta t) \end{bmatrix} = \begin{bmatrix} x_i(t) \\ y_i(t) \\ T_i^t(t) \end{bmatrix} + \begin{bmatrix} V_i^d \cdot \cos(\phi_i) \\ V_i^d \cdot \sin(\phi_i) \\ -\Delta t \end{bmatrix}
\tag{20}
$$

where $V_i^d = l_i(t_n^d)/KT_i(t_n^d)$ is the average speed of the decision at $t_n^d$. When the timer value of agent $i$ is less than zero (i.e., $T_i^t(t) < 0$), the agent $i$ comes into a new decision moment $t_{n+1}^d$ to ask the server for a new motion target. If the distance to the wall is less than one body length, the agent resets its timer to stop one kick process and then asks the Python server for a new decision (see Algorithm 1 for details).

---

**Algorithm 1.** States Update Rules for Agent $i$

---

**Input: decision results** $\begin{bmatrix} \delta\phi_i(t_n^d) & l_i(t_n^d) & KT_i(t_n^d) \end{bmatrix}^T$, **old states** $[x_i(t), y_i(t), \phi_i(t)]^T$ **and the timer value** $T_i^t(t)$.

**Output: new states** $[x_i(t+\Delta t), y_i(t+\Delta t), \phi_i(t+\Delta t)]^T$ **and new timer value** $T_i^t(t+\Delta t)$

If $T_i^t(t)$ is less than or equal to 0 \\ there exists a new decision from the Python server

$T_i^t(t+\Delta t)=KT_i(t_n^d)$

$\phi_i(t+\Delta t) = \phi_i(t) + \delta\phi_i(t_n^d)$

Else　　　　　　　　　　　　　\\ agent straight motion

$T_i^t(t+\Delta t)=T_i^t(t) - \Delta t$

$x_i(t+\Delta t)=x_i(t)+V_i^d \cdot \cos(\phi_i)$

$y_i(t+\Delta t)=y_i(t)+V_i^d \cdot \sin(\phi_i)$

If $r_i^w <$ BL　　　　\\ safety mechanism of the motion simulation

$T_i^t(t) = 0$　　　\\ ask the Python Server for a new decision

---

### 2.5. Statistical Properties of Collective Motion

Five agents' simulation trajectory results could be used for the comparison with the five-fish experiment to evaluate the effectiveness of our model. We selected six different statistical properties for collective motion. Consider the position of the barycenter (center of mass) $\vec{p}_B = (x_B(t), y_B(t))$, which is calculated with the following formulas:

$$
x_B(t) = \frac{1}{N}\sum_{i=1}^{N} x_i(t), \; y_B(t) = \frac{1}{N}\sum_{i=1}^{N} y_i(t)
\tag{21}
$$

where $N$ is the total number of agents in the group. Based on the position of the barycenter, the speed of the barycenter $(v_B^x(t), v_B^y(t))$ is defined by $v_B^x(t) = (x_B(t) - x_B(t-\Delta t))/\Delta t$ and $v_B^y(t) = (y_B(t) - y_B(t-\Delta t))/\Delta t$. Then, the direction of the barycenter is given by $\phi_B(t) = \arctan\left(v_B^y(t)/v_B^x(t)\right)$.

The barycenter holds a reference system in which the relative position and velocity of the fish are defined as $\vec{p}_i^B = \vec{p}_i - \vec{p}_B = (x_i^B(t), y_i^B(t))$ and $\vec{v}_i^B = \left((x_i^B(t) - x_i^B(t-\Delta t))/\Delta t, (y_i^B(t) - y_i^B(t-\Delta t))/\Delta t\right)$, respectively.

In the global reference system, we defined six statistical properties as follows:

1.　The distance from all fish (agents) to the wall: $r_W = \{r_i^W(t)|i = 1, 2, \cdots, N\}$;
2.　The angle to the wall of all fish (agents): $\theta_{W+} = \{|\theta_i^W(t)||i = 1, 2, \cdots, N\}$.

The above two statistical properties illustrate the position of the agents with respect to the global environment.

3. Polarization of the group: $P(t) \in [0,1]$:

$$P(t) = \frac{1}{N} \sum_{i=1}^{N} \vec{e}_i(t) \tag{22}$$

where $\vec{e}_i(t) = (\cos(\phi_i(t)), \sin(\phi_i(t)))$ is the unit vector for representing the direction of the fish $i$. When the value $P(t) = 1$, all the fish (agents) have the same orientation, while when this value is close to 0, all the fish are in different directions.

4. Group size: $C(t)$:

$$C(t) = \frac{1}{N} \sum_{i=1}^{N} \left\| \vec{p}_i - \vec{p}_B \right\| \tag{23}$$

where $\left\| \vec{p}_i - \vec{p}_B \right\|$ is the distance from $\vec{p}_i$ to $\vec{p}_B$. When the group is compact, $C(t)$ is low, and vice versa.

In the relative coordinate system with respect to the barycenter, we defined the following two characters for collective behavior:

5. Counter-milling index $Q(t) \in [-1,1]$:

$$Q(t) = \left( \frac{1}{N} \sum_{i=1}^{N} \sin\left(\gamma_i^B(t)\right) \right) \times \text{SIGN}\left( \frac{1}{N} \sum_{i=1}^{N} \sin\left(\theta_i^W(t)\right) \right) \tag{24}$$

where $\gamma_i^B(t)$ is the relative speed angle between the relative position vector $\vec{p}_i^B$ and the relative speed vector $\vec{v}_i^B$, $\text{SIGN}(\cdot)$ represents the sign function, and $\frac{1}{N} \sum_{i=1}^{N} \sin\left(\theta_i^W(t)\right) > 0$ means all fish (agents) move counterclockwise. Contrarily, if this value is less than zero, all fish move clockwise. When the direction of all fish moving around the center of the experimental space is different from the direction of each fish rotating around the barycenter, we call the group counter-milling swimming (i.e., $Q(t) < 0$). On the contrary, if the two directions of rotation are the same, then the group shows a super-milling behavior (i.e., $Q(t) > 0$).

6. The relative speed to the barycenter of all fish $\Delta V(t)$ may be described as follows:

$$\Delta V(t) = \left\{ \frac{\left\| \vec{p}_i(t) - \vec{p}_B(t) \right\| - \left\| \vec{p}_i(t - \Delta t) - \vec{p}_B(t - \Delta t) \right\|}{\Delta t}, i = 1, 2, \cdots, N \right\} \tag{25}$$

## 3. Results

The results are mainly divided into two parts. The first part is the analysis of pairwise interaction of the model in the two-fish (agent) experiment (simulation). The second part describes the analysis of the neighbor selection hypotheses in the five-fish (agent) experiment (simulation).

### 3.1. The Effect of Model Pairwise Interaction

In Figure 6, we present the distributions of six properties for the pairwise interaction behavior of the DNN model. These properties include three output values $[\delta\phi_1, l_1, KT_1]^{\mathrm{T}}$ and three statistical properties $[\Delta\phi_{12}^+(t), r_W, \theta_{W+}]^{\mathrm{T}}$. We input the experimental data $[r_1^W, \theta_1^W, V_1, d_{12}, \psi_{12}, \Delta\phi_{12}, \Delta V_{12}]^{\mathrm{T}}$ into the trained DNN model to check the consistency of the output value $[\delta\phi_1, l_1, KT_1]^{\mathrm{T}}$ (black lines) with respect to the fish decision (red lines) in Figure 6A–C. This can be regarded as the training error of the fish motion data.

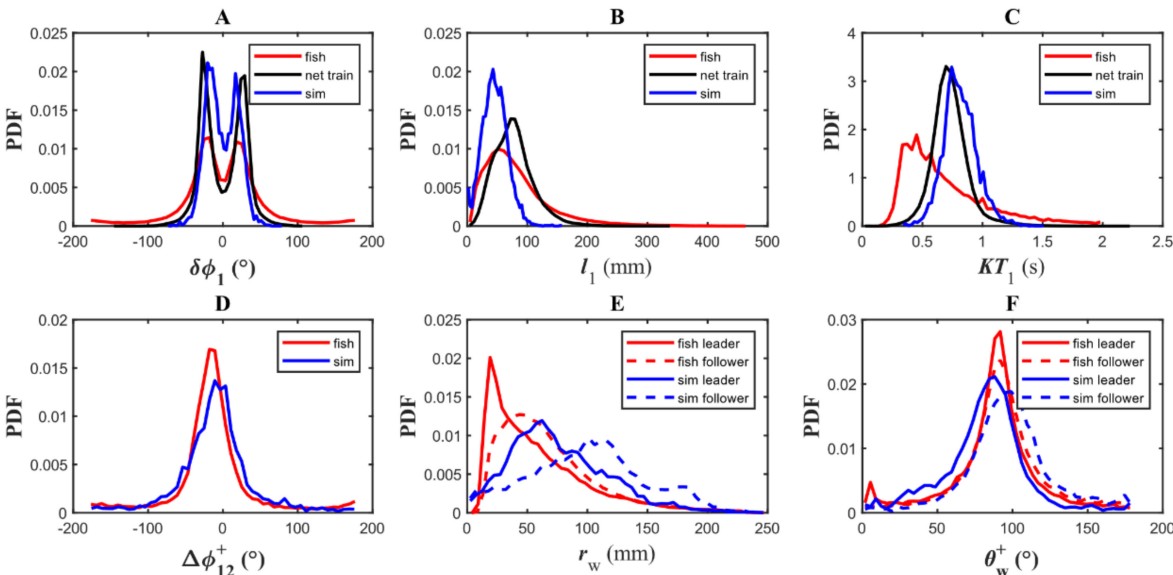

**Figure 6.** The comparison between the two-agent DNN model simulation and two–fish experiment. (**A**) PDF of the heading angle change after each kick of fish 1 $\delta\phi_1$; (**B**) PDF of the distance between two kicks of fish 1 $l_1$; (**C**) PDF of the duration between two kicks of fish 1 $KT_1$; (**D**) PDF of the signed relative angle of one individual between two fish $\Delta\phi_{12}^+(t) = \Delta\phi_{12} \times \text{SIGN}(\theta_1^W)$; (**E**) PDF of the distance to two fish. The solid lines represent the leaders. The dash lines represent followers. $r_W$; (**F**) PDF of the absolute value of the relative angle to the wall for the leader and follower $\theta_{W+}$.

Figure 6A–C shows that both the simulation (blue lines) and training data outputs (black lines) of the DNN were narrower than that of the real fish (red lines) because the DNN model could filter the noise of the original data of the fish schooling. Thus, if the DNN model is used in a simulation with training data input, the PDF of the output values indeed becomes narrower than the original training data label. In order to prevent overfitting, we stopped the learning iteration when the training error became larger. Hence, the DNN model could learn the general characteristics of the data, and it filtered out the special features such as the noise of the individual. As a result, the distribution of the DNN model output was sharper than that of the real fish. The peaks of the kick length for both the real fish and the DNN simulation were around 2BL (60 mm), and the average values of kick duration distribution of the DNN output, simulation, and real fish were similar.

We then compared the signed relative orientation $\Delta\phi_{12}^+(t) = \Delta\phi_{12} \times \text{SIGN}(\theta_1^W)$ between the two fish, which illustrated their direction of alignment (see Figure 6D). The simulation result showed that two agents aligned all the time that the real fish did. In Figure 6E, the solid and dashed lines show the relative distance to the wall of the leader and follower, respectively. We defined the leader as the agent (fish) with the larger the viewing angle, and thus $\left|\psi_{ij}(t)\right|$ was in the range of $[90°, 180°]$. Note that the literature [11] indicates that the leader and follower relationship is not stable in the experiment (i.e., fish change roles all the time).

In order to deeply analyze the pairwise interaction, we plotted the relationship between the heading angle change $\delta\phi_1$ and velocity change values (proportional to acceleration) of the focal individual 1 and the different relative speed $\Delta V_{12}$, position $(\Delta x_{12}, \Delta y_{12})$, and orientation $\Delta\phi_{12}$ of its neighbor, individual 2 (see Figure 7A,B). Since there were seven parameters for the input of the DNN pairwise interaction model $[r_1^W, \theta_1^W, V_1, d_{12}, \psi_{12}, \Delta\phi_{12}, \Delta V_{12}]^T$, we fixed $r_1^W$ at 100 mm and $\theta_1^W = -90°$. This means that the heading angle of the individual was always parallel to the wall. We selected these two parameter values because the fish group swam at this environmental position frequently (see Figure 6E,F). The focal fish swam clockwise when $\theta_1^W = -90°$. Hence, the wall was on its left side (see the left curve of Figure 7).

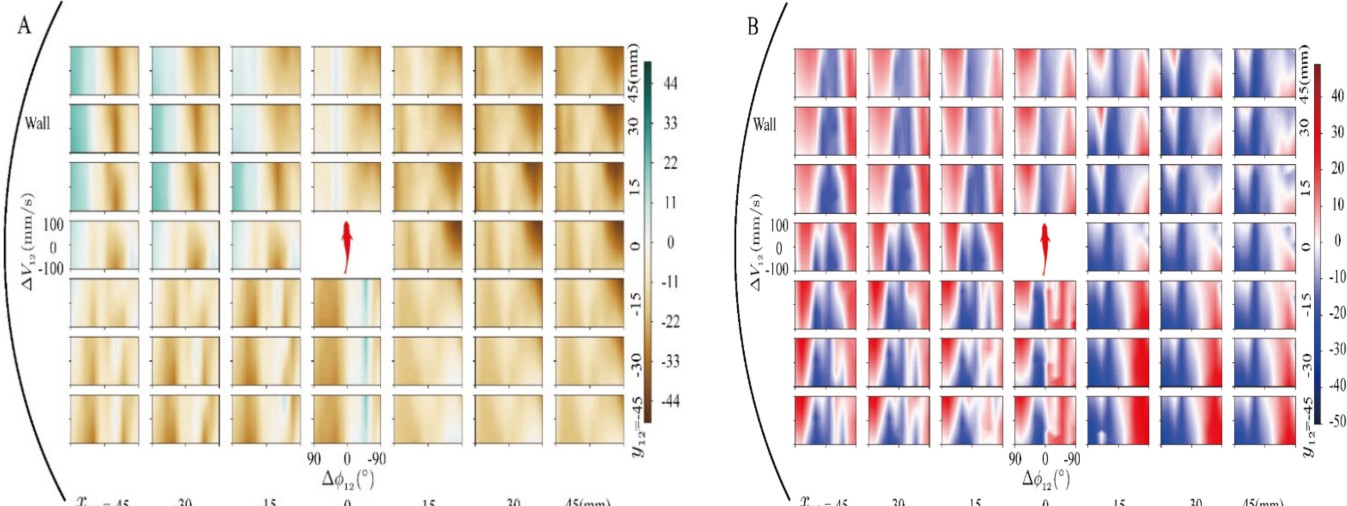

**Figure 7.** The relationship between the motion decision of the focal individual (red) and the relative position, orientation, and speed of its neighbor. (**A**) The relationship between the heading angle change $\delta\phi_{12}$ and perceptual information $[d_{12}\sin(\psi_{12}), d_{12}\cos(\psi_{12}), \Delta\phi_{12}, \Delta V_{12}]$ input of the ACN model. Each panel plots $\delta\phi_{12}$ with respect to $\Delta\phi_{12}$ and $\Delta V_{12}$ when the neighbor was located at $(\Delta x_{12}, \Delta y_{12}) = (d_{12}\sin(\psi_{12}), d_{12}\cos(\psi_{12}))$ in the range of 45 mm around the focal individual. Forty-eight different relative positions $(\Delta x_{12}, \Delta y_{12})$ of the neighbor in the Cartesian system are shown as subpanels. The speed of the focal fish was fixed at the median velocity of $V_1$ = 50 mm/s, and the environmental position of the focal fish was fixed at position $r_1^W$ = 100 mm and $\theta_1^W = -90°$. The green color means the focal individual turned left, while the yellow color means a right turn was made. (**B**) All of the structure is the same as in A, but all subpanels display the speed change of the focal fish $l_1/KT_1 - V_1$, which is related to the output of the LDN model. The red color (respectively. blue color) means the focal fish accelerated (respectively. decelerated) in the next kick. The x axis of each panel is the relative orientation of the neighbor, where a positive value means the neighbor's heading angle is on the left of the focal fish.

The velocity of the focal fish $V_1(t)$ was fixed at 50 mm/s so that the range of the decision speed could be extended to [0, 100 mm/s] in Figure 7B, which facilitated showing the speed change rules. We transformed the relative polar position of the neighbor $(d_{12}, \psi_{12})$ into a relative Cartesian coordinate $(x_{12}, y_{12})$ and plotted panels every 15 mm around the focal fish. The $x$ axis of each panel is the relative orientation of the neighbor, where a positive value means the neighbor's heading angle was on the left with respect to the focal individual. Therefore, we used an inverse $x$ axis coordinate in each panel for better visualization. The $y$ axis of each panel is the relative speed of the neighbor, ranging from [−100 mm/s, 100 mm/s].

The heading angle change $\delta\phi_1(t_n^d)$ of the focal fish caused by pairwise interaction is illustrated in Figure 7A. The green color means turning left, while yellow means turning right. Because of the left wall, the focal fish mainly turned right to avoid an environmental collision. This led to the main color of each panel in Figure 7A being yellow. We investigated the speed variation $l_1/KT_1 - V_1$ of the focal fish reflecting the influence of the neighbor, which was determined by the output of the LDN and the speed of the local fish (see Figure 7B). The red color means acceleration of the focal fish, while blue means deceleration.

The focal fish was sensitive to the heading angle of the front neighbor (see the strong alignment of the three top central panels of Figure 7A). If the neighbor's heading was on the left or right, the focal turned left or right (green and yellow). Furthermore, the focal fish decelerated for collision avoidance when its orientation was the same as that of the neighbor. On the other hand, the focal fish accelerated for neighbor attraction when the orientation of the neighbor was different from that of the focal (see the three top central panels of Figure 7B).

The bottom three central panels of Figure 7A,B show a situation where the neighbor was behind the focal fish. If the neighbor moved to the left, the focal fish would turn right

(yellow) and decelerate to wait for the neighbor. On the contrary, the focal fish turned left (green) to go outside of the tank with a speed and acceleration for keeping the leading position. Additionally, the focal fish kept its direction of motion when the neighbor had the same orientation.

If the neighbor was on the left of the focal fish, the focal fish decelerated when $\Delta\phi_{12} = 0$ and accelerated for attraction when $|\Delta\phi_{12}|$ was large. If the neighbor was on the right and the wall was on the left, the focal fish decelerated to prevent colliding with the wall. If the neighbor was on the left and in front, the focal fish inclined to align with the neighbor. Contrarily, if the neighbor was on the left and behind, the focal fish neglected it and decided to turn right to avoid the wall.

### 3.2. The Analysis of the Multi-Fusion Method of Pairwise Interaction

The cohesion of the group in the fish experiment (fish, red lines) and in the DNN model simulation was high, with C ≈ 50mm (see Figure 8D). However, the DNN model simulation with the Near Neighbor Selection (NNS, green lines) strategy was low. The cohesion PDF of the Random Neighbor Selection (RNS, blue lines) strategy of simulation was wider than that of the fish. Only the Large Visual Pressure Neighbor Selection (LVPS, black lines) strategy for the DNN model simulation was more compact than that of the fish. Figure 8C shows the PDF of polarization of the group. All the strategies were highly polarized, except that of NNS, which means that all individuals swam in the same direction (with a huge peak at P ≈ 1). All individuals swam near to the wall ($r_W$ ≈ 50mm) (see Figure 8A) and were always parallel to the wall ($\theta$ ≈ 90°) (see Figure 8B). The PDF of the DNN model simulation was sharper than that of the fish experiment, because the DNN model filtered the noise of the pairwise interaction. The relative speed shown in Figure 8F was similar for both the DNN model simulation and the fish experiment.

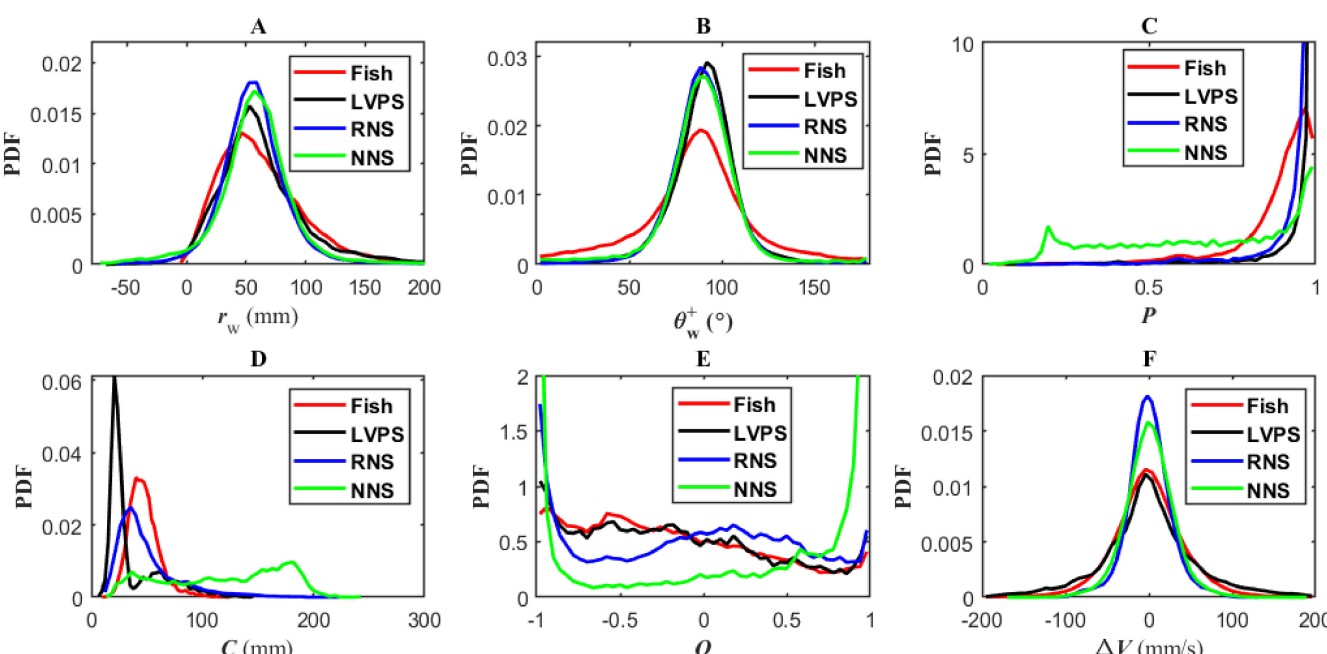

**Figure 8.** The comparison between the five-fish experiment and the five-agent DNN model simulation. (**A**) PDF of the relative distance from the wall $r_w$ of each individual; (**B**) PDF of the absolute value of relative angle to the wall $\theta_W^+$ of each individual; (**C**) PDF of the Polarization $P$; (**D**) PDF of the group cohesion $C$; (**E**) PDF of collective counter-milling and supper-milling index $Q$; (**F**) PDF of relative speed $\Delta V$ of the individual to the barycenter.

Counter-milling behavior was observed more frequently than over-milling behavior in the fish experiment (see Figure 8E). The counter-milling behavior was caused by the fact that the leader fish (at the front of the group) decelerated as they were closest to the

wall. On the contrary, the follower fish had more space to go inside of the wall, and hence they moved faster than the leader. Thus, the follower would catch up with the leader and become a new leader in front of the group. This operation was repeated continuously, causing all fish to rotate around the group barycenter. In counter-milling behavior, the direction of rotation around the barycenter was different with the direction of the group swimming around the experimental tank. This collective behavior was mainly caused by the asymmetric interaction.

The results of Figure 8 show that the LVPS strategy could lead to a more compact and stable collective motion than other neighbor selection strategies which was more similar to the real fish.

We extend our pairwise interaction model with LVPS strategy to the simulation with 100 agents (see Figure 9). It took about 2.5 min to aggregate a compact collective motion group from a random state. Compared with other social interaction models, our model could formulate collective motion by only interacting with one neighbor, which was selected by the LVPS strategy. This character allowed an individual to spend less computational load on formulating the collective motion.

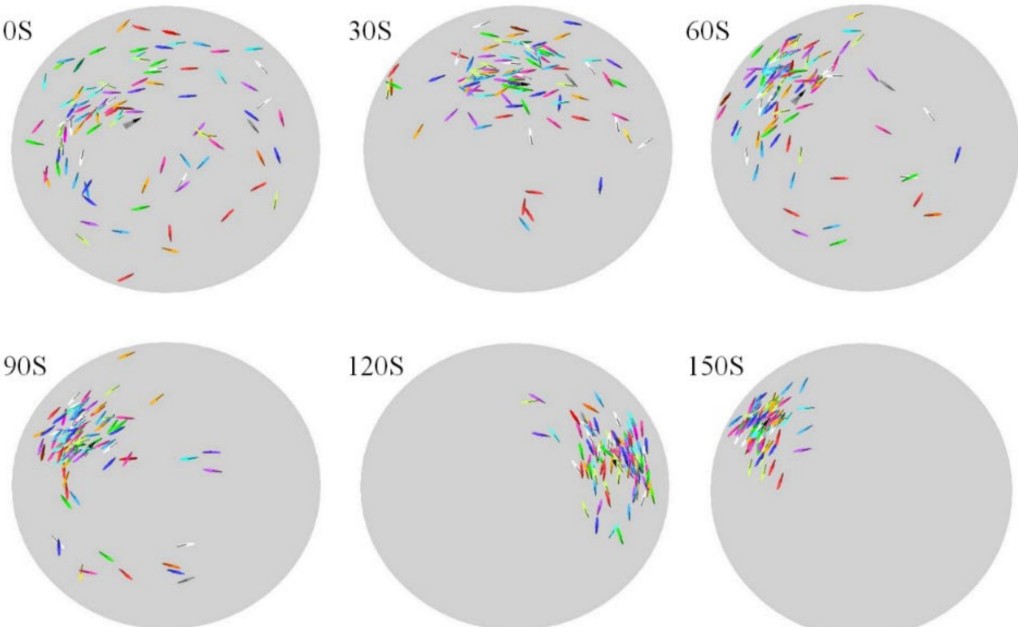

**Figure 9.** Six sequential frames of the 100-agent simulation for the aggregation of collective motion.

Finally, we applied the DNN model with the LVPS strategy on three "Cuboids" robots for collective motion (see Figure 10). The diameter of the circular robot platform was 1000 mm, and the body length and width of the Cuboid robot were both 40 mm. Figure 10 shows the control structure and related functions of the Cuboid robots' experimental platform; detailed information about the Cuboid robots' platform is illustrated in [32]. We spent one hour with three robots in a collective motion experiment. Figure 10A–E shows the top view sequence of the robots' motion in the experiment. The PDF of the Cuboid robot experiment is shown in Figure 11. All robots ran around the wall in the experiment. The relative distance to the wall was small (see Figure 11A). Figure 11B shows that the relative angle to the wall was kept at approximately 90 degrees. The polarization of the group was high, which was similar to that of the fish group (see Figure 11C). Meanwhile, Figure 11D shows that the cohesion of the group was also high, which meant that the Cuboid robot group was always compact.

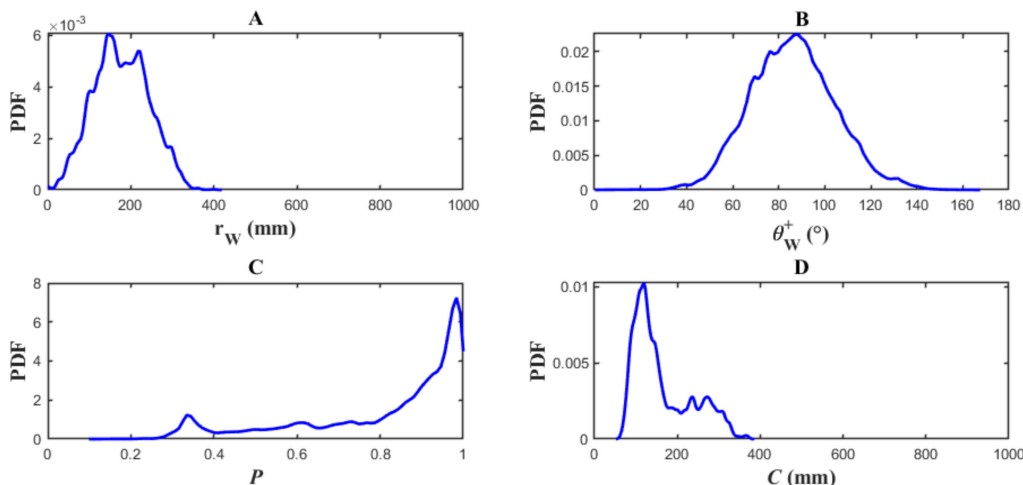

**Figure 10.** The Cuboid robots' platform and the top view sequence of the collective motion experiment.

**Figure 11.** The probability density function (PDF) of the three Cuboid robots collective motion experiment with the DNN model and LVPS strategy. (**A**) PDF of the distance from the wall to each robot $r_W$; (**B**) PDF of the absolute value of relative angle to the wall of each robot $\theta_W^+$; (**C**) PDF of the robot group polarization $P$; (**D**) PDF of the group cohesion $C$.

## 4. Discussion and Conclusions

In collective motion, each individual should adjust its behavior to adapt to its neighbors. Previous works suggest that one individual needs to interact with a lot of neighbors to achieve group cohesion [34–36]. However, focusing on one neighbor can overcome the low

information processing ability of the individual. Therefore, selecting the most influential neighbor in the group with which to interact is very important for understanding the coordination mechanisms of the group.

Here, we developed a pairwise interaction model of collective motion based on the DNN model, which could achieve stable collective motion with one-neighbor interaction. The neighbor was selected by the largest visual pressure. The results comparison between the two- and five-fish experiments and the DNN model simulation verified the motion similarity between our method and natural fish. All simulation agents perfectly had the same moving direction with the compact group. Counter-milling occurred in both fish groups and the agents' simulation with the LVPS strategy. This property of collective behavior enabled all individuals to alternate their positions in the group. For large-scale collective motion, we extended our method to 100 agents for simulation to verify the aggregation ability. The simulation showed that our method could formulate stable large-scale collective movement in a small period.

Compared with the deep attention network model proposed in [46], it can only provide the possibility of turning the direction of the individual. However, our model can output not only the specific steering angle, but also the straight moving distance and time. Pairwise interaction analysis showed that the follower individual preferred decelerating its speed for safety to maintain the alignment rather than change its heading angle for front neighbor avoidance, which was different from the results of other studies in the literature [11]. When the focal fish becomes the leader, it keeps its speed unchanged when the followers are aligning, whether the follower's speed is fast or slow. When the follower's orientation angle is different from that of the leader, the leader turns to maintain its leadership.

The proposed method has the potential to control swarm robotics. The performance of the Cuboid robots' motion was similar to that of the fish. This means that the DNN model control for real swarm robotics had stable, flexible, and scalable characters. This was because the fish's collective motion was robust and flexible. These good control characters can help swarm robotics to be applied in many areas, such as swarm robotic multi-robot cooperative pursuits [47] and exploration missions in dangerous areas with swarms [48].

Our pairwise interaction DNN model can integrate the information of both the static environment and the dynamic neighbors. Since this is only the primary research of deep learning technology in swarm robotic control, the model should have the ability to avoid collision by prediction. In the future, we will add predictive information of the neighbor to the deep network model and explore more complex environments which can handle the large-scale traffic congestion of swarm robotics.

**Author Contributions:** Conceptualization, L.L.; methodology, H.Z., L.L.; software, H.Z., L.L.; writing-original draft preparation, H.Z., L.L.; writing-review and editing, H.Z., L.L.; visualization, H.Z., L.L.; supervision, L.L.; project administration, L.L.; funding acquisition, L.L. All authors have read and agreed to the published version of the manuscript.

**Funding:** The research work and the APC was funded by the National Natural Science Foundation of China 72071130.

**Institutional Review Board Statement:** Not applicable.

**Informed Consent Statement:** Not applicable.

**Data Availability Statement:** No new data were created or analyzed in this study. Data sharing is not applicable to this article.

**Conflicts of Interest:** The authors declare no conflict of interest.

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
