# Peer review of "Intelligent Control of Swarm Robotics Employing Biomimetic Deep Learning"

_machines, doi:10.3390/machines9100236_

Round 1

Reviewer 1 Report

The paper presents a new model for collective motion based on pairwise interaction to reduce the computational cost for information processing of existing approaches.
The authors propose a new method for neighbour selection based on the largest visual pressure and use a DNN to determine the group motion model elements.

The problem addressed in the paper is clear and well-motivated. The presented results support the working hypothesis, demonstrating that pairwise interaction can also realise collective motion.
I have only some minor comments.
In section 2, the acronym PDF is used for the first time. Would you please define it here?

Figure 4 is helpful for the understanding of the proposed LVP strategy. I suggest including in the figure all the elements reported in equations 17 and 18, similarly to the previous Fig.1.

In section 3.1, the authors refer to six proprieties for the pairwise interaction in Fig 6 that refer to the elements of the DNN model. Since they are different from those previously cited in section 2, I suggest rephrasing the sentence to avoid misunderstandings.

In line 374, the authors state that simulation and training data outputs of DNN are narrower. But isn't that what we expect since the simulation uses the DNN?

Line 389. Authors claim that the leader and the follower relationship is not stable. This is intuitively understandable, but I don't understand how the instability is shown in Figure 6. Can authors better explain as such a situation can be interpreted from the figures?

Figure 8 have two sub-figures labelled with E. Please fix the second with F.

Line 459. I think that Fig8.E cited in line 459 should be Fig.8 F, as suggested before.

Finally, the layout of the paper needs some adjustments to improve its readability. I suggest, for example, reducing the captions of the figures (these explanations could be inserted in the text), adding space between the algorithm and the text of the section, adjusting spaces between lines and so on.

Reviewer 2 Report

 Thanks to the authors submit the paper named"Intelligent Control of Swarm Robotics Employing Biomimetic 2 Deep Learning". The author analyzed the collective motion on Hemigrammus Rhodostomus fish and built an interaction model based on the Deep Learning technique.  Based on the authors' previous work that individuals paying attention to only a few neighbors can lead to collective motion, the author further study and tests methods individuals only interact with one neighbor in this paper.

This manuscript is well written, the content is well-organized, the experiment is designed appropriately, the figures are clear and easy to understand, the results are comprehensive.

I have only one concern about the experiment. The author has investigated the fish's motion in a circle pool environment both in the real and simulated environment, compare to the real scene, this environment in this paper is a little bit simple. So if the author can conduct several experiments with obstacles in different environments, this paper will be even more comprehensive. 
